# A Flexible Framework for Designing Trainable Priors with Adaptive Smoothing and Game Encoding

**Bruno Lecouat**
Inria*
bruno.lecouat@inria.fr

**Jean Ponce**
Inria*
jean.ponce@inria.fr

**Julien Mairal**
Inria†
julien.mairal@inria.fr

## Abstract

We introduce a general framework for designing and training neural network layers whose forward passes can be interpreted as solving non-smooth convex optimization problems, and whose architectures are derived from an optimization algorithm. We focus on convex games, solved by local agents represented by the nodes of a graph and interacting through regularization functions. This approach is appealing for solving imaging problems, as it allows the use of classical image priors within deep models that are trainable end to end. The priors used in this presentation include variants of total variation, Laplacian regularization, bilateral filtering, sparse coding on learned dictionaries, and non-local self similarities. Our models are fully interpretable as well as parameter and data efficient. Our experiments demonstrate their effectiveness on a large diversity of tasks ranging from image denoising and compressed sensing for fMRI to dense stereo matching.

## 1 Introduction

Despite the undeniable successes of deep learning in domains as varied as image processing [63] and recognition [21], natural language processing [10], speech [43] or bioinformatics [1], feed-forward neural networks are often maligned as being "black boxes" that, except perhaps for their top classification or regression layers, are difficult or even impossible to interpret. In imaging applications, for example, the elementary operations typically consist of convolutions and pointwise nonlinearities, with many parameters adjusted by backpropagation, and no obvious functional interpretation.

In this paper, we consider instead network architectures explicitly derived from an optimization algorithm, and thus interpretable from a functional point of view. The first instance of this approach we are aware of is LISTA [20], which provides a fast approximation of sparse coding. Yet, we are not content to design an architecture that provides a fast approximation to a given optimization problem, but we also want to learn a data representation pertinent for the corresponding task. This yields an unusual machine learning paradigm, where one learns the parameters of a parametric objective function used to represent data, while designing an optimization algorithm to minimize it efficiently.

Even though interpretability is not always necessary to achieve good prediction, this point of view, sometimes called algorithm unrolling [17, 40], has proven successful for solving inverse imaging problems, providing effective and parameter-efficient models. This approach allows the use of domain-specific priors within trainable deep models, leading to a large number of applications such as compressive imaging [53, 62], demosaicking [26], denoising [26, 50, 52], and super-resolution [56] .

However, existing approaches are often limited to simple image priors such as sparsity induced by the $\ell_1$-norm [52], or differentiable regularization functions [27], and a general algorithmic framework

for combining complex, possibly non-smooth, regularization functions is still missing. Our paper addresses this issue and is able to leverage a large class of image priors such as total variation [49], the $\ell_1$-norm, structured sparse coding [34], or Laplacian regularization, where local optimization problems interact with each others. The interaction can be local among direct neighbors on an image grid, or non-local, capturing for instance similarities between spatially distant image patches [5, 12].

In this context, we adopt a more general and flexible point of view than the standard convex optimization paradigm, and consider formulations to represent data based on non-cooperative games [42] potentially involving non-smooth terms, which are tackled by using the Moreau-Yosida regularization technique [22, 61]. Unrolling the resulting optimization algorithm results in a network architecture that can be trained end-to-end and capture any combination of the domain-specific priors mentioned above. This approach includes and improves upon specific trainable sparse coding models based on the $\ell_1$-norm for example [52, 56]. More importantly perhaps, it can be used to construct several interesting new image priors: In particular, we show that a trainable variant of total variation and its non-local variant based on self similarities is competitive with the state of the art in imaging tasks, despite using up to 50 times fewer parameters, with corresponding gains in speed. We demonstrate the effectivness and the flexibility of our approach on several imaging tasks, namely denoising, compressed fMRI reconstruction, and stereo matching.

**Summary of our contributions.** First, we provide a new framework for building trainable variants of a large class of domain-specific image priors . Second, we show that several of these priors match or even outperform existing techniques that use a much larger number of parameters and training data. Finally, we present a set of practical tricks to make optimization-driven layers easy to train.

## 2   Background and Related Work

**Classical image priors.** Inverse imaging problems are often solved by minimizing a data fitting term with respect to model parameters, regularized with a penalty that encourages solutions with a particular structure. In image processing, the community long focused on designing handcrafted priors such as sparse coding on learned dictionaries [14, 33], diffusion operators [45], total variation [49], and non-local self similarities [5], which is a key ingredient of successful restoration algorithms such as BM3D [12]. However these methods are now often outperformed by deep learning models [29, 63, 64], which leverage pairs of corrupted/clean training images in a supervised fashion.

**Bilevel optimization.** A simple method for mixing data representation learning with optimization is to use a bi-level formulation [32]. For instance, assuming that one is given pairs $(\mathbf{x}_i, \mathbf{y}_i)_{i=1...n}$ of corrupted/clean signals with $\mathbf{x}_i$ and $\mathbf{y}_i$ in $\mathbb{R}^m$, one may consider the following bi-level objective

$$\min_{\boldsymbol{\theta} \in \Theta, \mathbf{W} \in \mathbb{R}^{m \times p}} \frac{1}{n} \sum_{i=1}^n L(\mathbf{y}_i, \mathbf{W}\mathbf{z}_{\boldsymbol{\theta}}^\star(\mathbf{x}_i)) \quad \text{where} \quad \mathbf{z}_{\boldsymbol{\theta}}^\star(\mathbf{x}_i) \in \arg\min_{\mathbf{z} \in \mathbb{R}^p} h_{\boldsymbol{\theta}}(\mathbf{x}_i, \mathbf{z}), \tag{1}$$

where $\boldsymbol{\theta}$ is a set of model parameters, $\mathbf{W}\mathbf{z}_{\boldsymbol{\theta}}^\star(\mathbf{x}_i)$ is a prediction which is compared to $\mathbf{y}_i$ through a loss function $L : \mathbb{R}^m \times \mathbb{R}^m \to \mathbb{R}^+$, and the data representation $\mathbf{z}_{\boldsymbol{\theta}}^\star(\mathbf{x}_i)$ in $\mathbb{R}^p$ is obtained by minimizing some function $h_{\boldsymbol{\theta}}$. Note that for simplicity, we have considered here a multivariate regression problem, where given a signal $\mathbf{x}$ in $\mathbb{R}^m$, we want to predict another signal $\mathbf{y}$ in $\mathbb{R}^m$, but this formulation also applies to classification problems. It was first introduced for sparse coding in [32, 58] and it has recently been extended to the case when $\mathbf{z}_{\boldsymbol{\theta}}^\star(\mathbf{x}_i)$ is replaced by an approximate minimizer of $h_{\boldsymbol{\theta}}$.

**Unrolled algorithms.** A common approach to solving (1) consists in choosing an iterative method for minimizing $h_{\boldsymbol{\theta}}$ and then define $\mathbf{z}_{\boldsymbol{\theta}}^\star(\mathbf{x}_i)$ as the output of the optimization method after $K$ iterations. The sequence of operations performed by the optimization method can be seen as a computational graph and $\nabla_{\boldsymbol{\theta}} \mathbf{z}_{\boldsymbol{\theta}}^\star$ can be computed by automatic differentiation. This often yields neural-network-like computational graphs, which we call *optimization-driven layers*. Such architectures have found multiple applications such as training of conditional random fields [65], stabilization of generative adversarial networks [38], structured prediction [3], or hyper-parameters tuning [31]. For image restoration, various optimization problems have been explored including for example sparse coding [26, 52, 62], non linear diffusion [9] and differential operator regularization [27]. Many inference algorithms have been investigated including proximal gradient descent [27, 52], ADMM [17], half quadratic splitting [66], or augmented Lagrangian [48].

# 3 A General Framework for Learning Optimization-Driven Layers

## 3.1 Proposed Approach

We adopt a more general point of view than (1), where we assume that input signals admit a local "patch" structure (*e.g.*, rectangular image regions) and the data representation encodes individual patches. Assuming that there are $m$ patches in $\mathbf{x}$, we denote by $\mathbf{Z}^\star(\mathbf{x}) = [\mathbf{z}_1^\star(\mathbf{x}), \ldots, \mathbf{z}_m^\star(\mathbf{x})]$ in $\mathbb{R}^{p \times m}$ the representation of $\mathbf{x}$ and by $\mathbf{z}_j^\star(\mathbf{x})$ the representation of patch $j$ (we omit the dependency on the model parameters $\boldsymbol{\theta}$ for simplicity). In imaging applications and as in previous models [52], $\mathbf{Z}^\star(\mathbf{x})$ can be seen as a feature map akin to that of a convolutional neural network with $p$ channels.

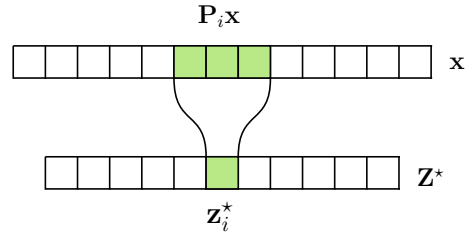

Figure 1: Our models encode locally an input feature vector. The local optimal solutions $\mathbf{z}_j^\star$ interact trough the regularization function $\psi_\theta^j(\mathbf{Z})$.

**Encoding with non-cooperative convex games.** Concretely, given a signal $\mathbf{x}$, we denote by $\mathbf{P}_j\mathbf{x}$ the patch of $\mathbf{x}$ centered at position $j$, where $\mathbf{P}_j$ is a linear patch extraction operator, and we define the optimal encoding $\mathbf{Z}^\star(\mathbf{x})$ of $\mathbf{x}$ as a Nash equilibrium of the set of problems

$$\min_{\mathbf{z}_j \in \mathcal{Z}} h_{\boldsymbol{\theta}}(\mathbf{P}_j\mathbf{x}, \mathbf{z}_j) + \psi_{\boldsymbol{\theta}}^j(\mathbf{Z}) \quad \text{for } j = 1, \ldots, m, \tag{2}$$

where $h_{\boldsymbol{\theta}}$ is a a convex reconstruction objective for each patch, parametrized by $\boldsymbol{\theta}$, $\psi_{\boldsymbol{\theta}}^j$ is a convex regularization function encoding interactions between the variable $\mathbf{z}_j$ and the remaining ones $\mathbf{z}_l$ for $l \neq j$, and $\mathcal{Z}$ is a convex subset of $\mathbb{R}^p$. When $\mathcal{Z}$ is compact, the problem is a specific instance of a non-cooperative convex game [42], which is known to admit at least one Nash equilibrium—that is, a solution such that one of the objectives in (2) is optimal with respect to its variable $\mathbf{z}_j$ when the other variables $\mathbf{z}_l$ for $l \neq j$ are fixed. The conditions under which an optimization algorithm is guaranteed to return such an equilibrium point are well studied, see Section 3.3, and in many situations the compactness of $\mathcal{Z}$ is not required, as also observed in our experiments where we choose $\mathcal{Z} = \mathbb{R}^p$. For instance, in several practical cases, (2) can be solved by minimizing the sum of $m$ convex terms, a setting called a *potential game*, which boils down to a classical convex optimization problem.

## 3.2 Application of our Framework to Inverse Imaging Problems

In this section, we show how to leverage our optimization-driven layers for imaging. For the sake of clarity we choose to narrow down the scope of this presentation to imaging, even though our method is not limited to this single application: different modalities including for example genomic/graph data could benefit from our methodology.

**Examples of models $h_{\boldsymbol{\theta}}$.** We consider two cases in the rest of this presentation:

- *Pixel reconstruction*: $h_{\boldsymbol{\theta}}(\mathbf{P}_j\mathbf{x}, \mathbf{z}_j) = (\mathbf{x}_j - \mathbf{z}_j)^2$, where $\mathbf{x}_j$ is the pixel $j$ of $\mathbf{x}$ and $\mathbf{z}_j$ is a scalar, corresponding to patches of size $q = 1 \times 1$ and $p = 1$.
- *Patch encoding on a dictionary*: $h_{\boldsymbol{\theta}}(\mathbf{P}_j\mathbf{x}, \mathbf{z}_j) = \|\mathbf{P}_j\mathbf{x}_j - \mathbf{D}\mathbf{z}_j\|^2$, where $\mathbf{D}$ in $\mathbb{R}^{q \times p}$ is a dictionary, $q$ is the patch size, and $p$ is the number of dictionary elements. This is a classical model where patch $j$ is approximated by a linear, often sparse, combination of dictionary elements [14].

Only the second choice involves model parameters $\mathbf{D}$ (represented by $\boldsymbol{\theta}$). These two loss functions are common in image processing [14], but other losses may be used for other modalities.

**Linear reconstruction with a dictionary.** Assuming that $\mathbf{y}$ and $\mathbf{x}$ have the same size $m$ for simplicity, predicting $\mathbf{y}$ from a feature map $\mathbf{Z}^\star(\mathbf{x})$ is typically achieved by using a learned dictionary matrix $\mathbf{W}$ in $\mathbb{R}^{q \times p}$ where $q$ is the patch size. Then, $\mathbf{W}\mathbf{z}_j^\star(\mathbf{x})^3$ can be interpreted as a reconstruction of the $j$-patch of $\mathbf{y}$. Since the patches overlap, we obtain $q$ estimators for every pixel, which can be combined by averaging (neglecting border effects below for simplicity), yielding the prediction

$$\hat{\mathbf{y}}(\mathbf{x}, \theta, \mathbf{W}) = \frac{1}{q} \sum_{j=1}^m \mathbf{P}_j^\top \mathbf{W} \mathbf{z}_j^\star(\mathbf{x}), \tag{3}$$

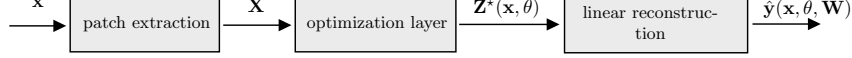

Figure 2: Architecture of our trainable models for image restoration.

where $\mathbf{P}_j^\top$ is the linear operator that places a patch of size $q$ at position $j$ in a signal of dimension $m$. Patch averaging is a classical operation in patch-based image restoration algorithms, see [14], which can be interpreted in terms of transposed convolution[4] and admits fast implementations on GPUs.

**Learning problem.** For image restoration, given training pairs of corrupted/clean images $\{\mathbf{x}_i, \mathbf{y}_i\}_{i=1,\ldots,n}$, we consider the regression problem $\min_{\theta \in \Theta, \mathbf{W} \in \mathbb{R}^{q \times p}} \|\mathbf{y}_i - \hat{\mathbf{y}}(\mathbf{x}_i, \theta, \mathbf{W})\|^2$ where $\hat{\mathbf{y}}(\mathbf{x}_i)$ is defined in (3).

**Examples of regularization functions $\psi_\theta^j$.** Our framework allows the use of several regularization functions, which are presented in the table 1 below. We assume that the patches are nodes in a graph, and denote by $\mathcal{N}_j$ the set of neighbors of the patch $j$. For natural images, the graph may be a two-dimensional grid with edge weights $a_{j,k}$ that depend on the relative position of the patches $j$ and $k$, which we denote by $a_{j-k}$, but it may also be a non-local graph based on some similarity function as in [26, 29]. Concretely, we can consider:

- the distance $d_{\mathrm{NL}}^{j,k} = \|\operatorname{diag}(\boldsymbol{\kappa})(\mathbf{P}_j\mathbf{x} - \mathbf{P}_k\mathbf{x})\|^2$ between patches $j$ and $k$ of the image $\mathbf{x}$, where $\boldsymbol{\kappa}$ in $\mathbb{R}^q$ is a set of parameters to learn, and $q$ is the patch size, and we define normalized weights $a_{\mathrm{NL}}^{j,k} = e^{-d_{\mathrm{NL}}^{j,k}} / \sum_{l \in \mathcal{N}_j} e^{-d_{\mathrm{NL}}^{j,k}}$.
- or a distance inspired from the bilateral filter [54]. In that case we define the distance $d_{\mathrm{BL}}^{j,k} = \frac{\|\mathbf{x}_i - \mathbf{x}_j\|^2}{2\sigma_d^2} + \frac{\|i - j\|^2}{2\sigma_r^2}$ between pixels on a local window $\mathcal{N}_j$ centered around pixel $j$, and we define again normalized weights $a_{\mathrm{BL}}^{j,k} = e^{-d_{\mathrm{BL}}^{j,k}} / \sum_{l \in \mathcal{N}_j} e^{-d_{\mathrm{BL}}^{j,k}}$.

Table 1: A non-exhaustive list of regularization functions $\psi_\theta$ covered by our framework.

|  | $\psi_\theta^j(\mathbf{Z})$ | Model parameters |
|---|---|---|
| Laplacian | $\sum_{k \in \mathcal{N}_j} a_{j-k} \|\mathbf{z}_j - \mathbf{z}_k\|^2$ | weights in $\mathbb{R}^{|\mathcal{N}|}$ |
| Non-local Laplacian | $\sum_{k \in \mathcal{N}_j} a_{\mathrm{NL}}^{j,k} \|\mathbf{z}_j - \mathbf{z}_k\|^2$ | $\boldsymbol{\kappa}$ in $\mathbb{R}^q$ |
| Bilateral filter (BF) | $\sum_{k \in \mathcal{N}_j} a_{\mathrm{BL}}^{j,k} \|\mathbf{z}_j - \mathbf{z}_k\|^2$ | $\sigma_d \in \mathbb{R}$ and $\sigma_r \in \mathbb{R}$ |
| Total variation (TV) | $\sum_{k \in \mathcal{N}_j} a_{j-k} \|\mathbf{z}_j - \mathbf{z}_k\|_1$ | weights in $\mathbb{R}^{|\mathcal{N}|}$ |
| Non-local total variation (NLTV) | $\sum_{k \in \mathcal{N}_j} a_{\mathrm{NL}}^{j,k} \|\mathbf{z}_j - \mathbf{z}_k\|_1$ | $\boldsymbol{\kappa}$ in $\mathbb{R}^q$ |
| Bilateral TV (BLTV) | $\sum_{k \in \mathcal{N}_j} a_{\mathrm{BL}}^{j,k} \|\mathbf{z}_j - \mathbf{z}_k\|_1$ | $\sigma_d \in \mathbb{R}$ and $\sigma_r \in \mathbb{R}$ |
| Weighted $\ell_1$-norm (sparse coding) | $\sum_{l=1}^p \lambda_l |\mathbf{z}_j[l]|$ | $\boldsymbol{\lambda}$ in $\mathbb{R}^p$ |
| Non-local group regularization | $\sum_{l=1}^p \lambda_l \sqrt{\sum_{k \in \mathcal{N}_j} a_{j,k} \mathbf{z}_k[l]^2}$ | $\boldsymbol{\lambda}$ in $\mathbb{R}^p$ and $\boldsymbol{\kappa}$ in $\mathbb{R}^q$ |
| Variance reduction | $\|\mathbf{W}\mathbf{z}_j - \mathbf{P}_j\hat{\mathbf{y}}\|^2$ with $\hat{\mathbf{y}}$ from (3) | $\mathbf{W}$ from (3) |

s

**Novelty of the proposed formulation and relation to previous work.**
- *Total variation*: to the best of our knowledge, the basic anisotropic TV penalty [6] does not seem to appear in the literature on unrolled algorithms with end-to-end training. Note also that our TV variant allows learning non-symmetric weights $a_{j,k} \neq a_{k,j}$, leading to a non-cooperative game that goes beyond the classical convex optimization framework typically used with the TV penalty.
- *Non-local TV*: the non-local TV penalty presented above is based on a classical formulation [19], but can be incorporated within a trainable deep network with non-symmetric weights.
- *Bilateral filtering*: the bilateral filter and its TV variant implemented in this paper are based on classical formulations [54, 16]. But they have not, to the best of our knowledge, been implemented as trainable priors.

- *Sparse coding and variance reduction*: the weighted $\ell_1$-norm combined with the patch encoding loss $h_{\boldsymbol{\theta}}$ yields a sparse coding formulation (SC) that has been well studied within optimization-driven layers [50, 52]. Yet, the codes $\mathbf{z}_j$ in the SC setup are obtained by solving independent optimization problems, which has motivated by Simon and Elad [52] to propose instead a Convolutional Sparse Coding model (CSC), where the full image is approximated by a linear combination of small dictionary elements. Unfortunately, as noted in [52], CSC leads to ill-conditioned optimization problems, making a hybrid approach between SC and CSC more effective. Our paper proposes an alternative solution combining the weighted $\ell_1$-norm regularization with a variance reduction penalty, which forces the codes $\mathbf{z}_j$ to reach a consensus when reconstructing the image $\hat{\mathbf{y}}$. Our experiments show that this approach outperforms [52] for image denoising.
- *Non-local group regularization*: This regularization function corresponds to a soft variant of the Group Lasso penalty [55], which encourages similar patches to share similar sparsity patterns (set of non-zero elements of the codes $\mathbf{z}_j$). It was originally used in [34] and was recently revisited within optimization-driven layers with an heuristic algorithm [26]. Our paper provides a better justified algorithmic framework as well as the ability to combine this penalty with other ones.

---

**Algorithm 1** Pseudocode of the general training procedure for image restoration

---

1: Sample a minibatch of pairs of corrupted/clean images $\{(\mathbf{x}_0, \mathbf{y}_0), \cdots, (\mathbf{x}_K, \mathbf{y}_K)\}$;
2: Extract overlapping patches of corrupted images to form tensors $\mathbf{X}_i = [\mathbf{P}_1\mathbf{x}_i, \cdots, \mathbf{P}_n\mathbf{x}_i]$;
3: **for** $t = 1, 2, \ldots, K$ **do**  $\quad\triangleright$ Compute an approximate Nash equilibrium $\mathbf{Z}^\star$ of the convex games
4: $\quad$ $\mathbf{Z}_{t+1} \leftarrow \mathbf{Z}_t - \eta_t H_\theta(\mathbf{Z}_t, \mathbf{X})$;
5: **end for**
6: Approximate clean images by linear reconstruction $\hat{\mathbf{y}} = \frac{1}{q} \sum_{j=1}^m \mathbf{P}_j^\top \mathbf{W} \mathbf{z}_j^\star(\mathbf{X}, \theta)$;
7: Compute the $\ell_2$ reconstruction loss $\|\mathbf{y} - \hat{\mathbf{y}}(\mathbf{x}, \theta, \mathbf{W})\|_2^2$ on the minibatch;
8: Compute an estimate of the gradients wrt. $(\theta, \mathbf{W})$ with auto-diff;
9: Update trainable parameters $(\theta, \mathbf{W})$ with Adam;

---

### 3.3 Differentiability and End-to-end Training

In this section we adress end-to-end training of the optimization-driven layers. Given pairs of training data $\{\mathbf{x}_i, \mathbf{y}_i\}_{i=1,\ldots,n}$, we consider the learning problem

$$\min_{\boldsymbol{\theta} \in \Theta} \frac{1}{n} \sum_{i=1}^n L\left(\mathbf{y}_i, g_\theta\left(\mathbf{z}_{\boldsymbol{\theta}}^\star(\mathbf{x}_i)\right)\right), \tag{4}$$

where $g_\theta$ is a differentiable function. We consider the approximation where the codes $\mathbf{z}_j^\star(\mathbf{x})$ are obtained as the $K$-th step of an optimization algorithm for solving the problem (2). To obtain these codes, we leverage (i) iterative gradient and extra-gradient methods, which are classical for solving game problems [4, 15], and (ii) a smoothing technique for dealing with the regularization functions $\psi_{\boldsymbol{\theta}}^j$ above when they are non-smooth. Refer to Algorithm 1 for an overview of the training procedure. We start with the first point when dealing with smooth objectives.

**Unrolled optimization for convex games.** Consider a set of $m$ objective functions of the form

$$\min_{\mathbf{z}_j \in \mathbb{R}^p} h_j(\mathbf{Z}) \quad \text{with} \quad \mathbf{Z} = [\mathbf{z}_1, \ldots, \mathbf{z}_m], \tag{5}$$

where the functions $h_j$ are convex and differentiable and may depend on other parameters than $\mathbf{z}_j$. Our objective is to find a zero of the simultaneous gradient

$$H(\mathbf{Z}) = [\nabla_{\mathbf{z}_1} h_1(\mathbf{Z}), \cdots, \nabla_{\mathbf{z}_m} h_m(\mathbf{Z})], \tag{6}$$

which corresponds to a Nash equilibrium of the game (5). In the rest of this presentation, we consider both the general setting and the simpler case of so-called potential games, for which the equilibrium can be found as the optimum of a single convex objective. This is the case for several of our regularizers, for example the TV penalty with symmetric weights. More details are provided in Appendix A on the nature of the non-cooperative games corresponding to our penalties.

Two standard methods studied in the variational inequality litterature [4, 15, 23, 37] are the *gradient* and the *extra-gradient* [25] methods. The iterates of the basic gradient method are given by

$$\mathbf{Z}_{t+1} = \mathbf{Z}_t - \eta_t H(\mathbf{Z}_t), \tag{7}$$

Table 2: Gradient descent (GD) vs. Extra-gradient. Denoising results in avg. PSNR with $\sigma = 25$ on BSD68 [35].

| Method | GD (24 iters) | GD (48 iters) | Extra-gradient (24 iters) |
|---|---|---|---|
| Trainable TV *symmetric* | 27.58 | 27.50 | 27.82 |
| Trainable TV *assymetric* | 27.99 | 27.89 | 28.24 |

where $\eta_t > 0$ is a step-size. These iterates are known to converge under a condition called strong monotonicity of the operator $H$, which is related to the concept of strong convexity in optimization, see [4]. Because this condition is relatively stringent, the extra-gradient method is often prefered [25], as it is known to converge under weaker conditions, see [23, 37]. The intuition of the method is to compute a look ahead step in order to compute more stable directions of descent:

$$
\begin{aligned}
\text{Extrapolation step} \quad & \mathbf{Z}_{t+1/2} = \mathbf{Z}_t - \eta_t H(\mathbf{Z}_t) \\
\text{Update step} \quad & \mathbf{Z}_{t+1} = \mathbf{Z}_t - \eta_t H(\mathbf{Z}_{t+1/2}).
\end{aligned}
\tag{8}
$$

In this paper, our strategy is to unroll iterates of one of these two algorithms, and then to use auto differentiation for learning the model parameters $\boldsymbol{\theta}$. Furthermore, parameters that control the optimization process (*e.g.*, step size $\eta_t$) can also be learnt with this approach. It should be noted that optimization-driven layers have never been used before in the context of non-cooperative games, to the best of our knowledge, and therefore an empirical study is needed to choose between the strategies (7) or (8). In our experiments, extra-gradient descent has always performed at least as well, and sometimes significantly better, than plain gradient descent for comparable computational budgets. See for example Table 2 for a smoothed variant of the TV penalty.

**Moreau-Yosida smoothing.** The non-smooth regularization functions we consider can be written as a sum of simple terms. Omitting the dependency on $\boldsymbol{\theta}$ for simplicity, we may indeed write

$$
\psi^j(\mathbf{Z}) = \sum_{k=1}^r \phi_k(L_{k,j}(\mathbf{Z})) \quad \text{for some } r \geq 1,
$$

where $L_{k,j}$ is a linear mapping and $\phi_k$ is either the $\ell_1$- or $\ell_2$-norm. For instance, $\phi_k$ is the $\ell_1$-norm with $L_{k,j}(\mathbf{Z}) = a_{k,j}(\mathbf{z}_j - \mathbf{z}_k)$ in $\mathbb{R}^p$ for the TV penalty, and $L_{k,j}(\mathbf{Z}) = [\sqrt{a_{1,j}}\mathbf{z}_1(k), \ldots, \sqrt{a_{1,j}}\mathbf{z}_q(k)]^\top$ in $\mathbb{R}^q$ with $\phi_k$ being the $\ell_2$-norm for the non-local group regularization. Handling such non-smooth convex terms may be achieved by leveraging the so-called Moreau-Yosida regularization [28, 41, 60]

$$
\Phi_k(\mathbf{u}) = \min_{\mathbf{v}} \left\{ \phi_k(\mathbf{v}) + \frac{\alpha}{2} \|\mathbf{v} - \mathbf{u}\|^2 \right\},
$$

which defines an optimization problem whose solution is called the proximal operator $\text{Prox}_{\phi_k/\alpha}[\mathbf{u}]$. As shown in [28], $\Phi_k$ is always differentiable and $\nabla \Phi_k(\mathbf{u}) = \alpha(\mathbf{u} - \text{Prox}_{\phi_k/\alpha}[\mathbf{u}])$, which can be computed in closed form when $\phi_k = \ell_1$ or $\ell_2$. The positive parameter $\alpha$ controls the trade-off between smoothness (the gradient of $\Phi_k$ is $\alpha$-Lipschitz) and the quality of approximation. It is thus natural to define a smoothed approximation $\Psi^j$ of $\psi^j$ as $\Psi^j(\mathbf{Z}) = \sum_{k=1}^r \Phi_k(L_{k,j}(\mathbf{Z}))$.

Note that when the proximal operator of $\psi^j$ can be computed efficiently, as is the case for the $\ell_1$-norm, gradient descent algorithms can typically be adapted to handle the non-smooth penalty without extra computational cost [33], and there is no need for Moreau-Yosida smoothing. However, the proximal operator of the TV penalty and the non-local group regularization do not admit fast implementations. For the first one, computing the proximal operator requires solving a network flow problem [7], whereas the second one is essentially easy to solve when the weights $a_{j,k}$ form non-overlapping groups of variables, leading to a penalty called group Lasso [55].

We are now ready to present our unrolled algorithm as we have previously discussed gradient-based algorithms for solving convex smooth games and a smoothing technique for handling non-smooth terms. Generally, at iteration $t$, the gradient algorithm (7) performs the following simultaneous updates for all problems $j$

$$
\mathbf{u}_{k,j}^{(t)} \leftarrow \text{Prox}_{\phi_k/\alpha_{k,t}}[L_{k,j}(\mathbf{Z}^{(t)})] \quad \text{for } k = 1, \ldots, r
$$

$$
\mathbf{z}_j^{(t+1)} \leftarrow \mathbf{z}_j^{(t)} - \eta_t \left( \nabla_{\mathbf{z}_j} h_\theta \left( \mathbf{P}_j \mathbf{x}, \mathbf{z}_j^{(t)} \right) + \sum_{k=1}^r \alpha_{k,t} \left[ L_{k,j}^* \left( L_{k,j}(\mathbf{Z}^{(t)}) - \mathbf{u}_{k,j}^{(t)} \right) \right]_j \right),
$$

where $L_{k,j}^*$ is the adjoint of the linear mapping $L_{k,j}$. The computation of the gradients can be implemented with simple operations allowing auto-differentiation in deep learning frameworks. Interestingly, the smoothing parameter $\alpha$ can be made iteration-dependent, and learned along with other model parameters such that the amount of smoothing is chosen automatically.

### 3.4 Tricks of the Trade for Unrolled Optimization

Our strategy is to unroll iterates of our algorithms, and then compute $\nabla_{\boldsymbol{\theta}} \mathbf{z}_{\boldsymbol{\theta}}^{\star}$ by automatic differentiation. We present here a set of practical rules, some old and some new, facilitating training when $h_{\boldsymbol{\theta}}$ is a patch encoding function on a dictionary $\mathbf{D}$.

**Initialization.** To help the algorithm converge, we choose an initial stepsize $\eta_t \leq \frac{1}{L}$, where $L$ is the Lipschitz constant of $\nabla_{\mathbf{z}} h_{\boldsymbol{\theta}}$, which is the classical step-size used by ISTA [2]. To do so, inspired by [52] we normalize the initial dictionary by its largest singular value and take $\eta_0 = 1$. Note that we can go one step further and normalize the dictionary throughout the training phase. This is in fact equivalent to the spectral normalization that has received some attention recently, notably for generative adversarial networks [39].

**Untied parameters.** In our framework, $\nabla_{\mathbf{z}_j} h_{\boldsymbol{\theta}}(\mathbf{P}_j \mathbf{x}, \mathbf{z}_j) = \mathbf{D}^{\top}(\mathbf{D}\mathbf{z}_j - \mathbf{P}_j \mathbf{x})$. It has been suggested in previous work [20, 26, 52] to introduce an additional parameter $\mathbf{C}$ of the same size as $\mathbf{D}$, and consider instead the parametrization $\mathbf{C}^{\top}(\mathbf{D}\mathbf{z}_j - \mathbf{P}_j \mathbf{x})$, $\mathbf{C}$ acting as a learned preconditioner. Even though the theoretical effect of this modification is not fully understood, it has been observed to accelerate convergence and boost performance for denoising tasks [26]. In our experiments, we will indicate in which cases we use this heuristic.

**Backtracking.** A simple way for handling the potential instability of the unrolled algorithm is to use a backtracking scheme which automatically decreases the stepsize when the training loss diverges. This heuristic was used for instance in [26]. More details are provided in Appendix B.

**Barzilai-Borwein method for choosing the stepsize.** A different, perhaps more principled, approach to improved stability consists in adaptively choosing an adaptive stepsize $\eta_t$. The literature on convex optimization proposes a set of effective rules, known as Barzilai-Borwein (BB) step size rules [57]. Even though these rules were not designed for convex games, they appear to be very effective in practice in the context of our optimization-driven layers. Concretely, they lead to step sizes $\eta_{t,j} = \|\mathbf{D}^{\top}\mathbf{D}\mathbf{s}_j\|_2 / \|\mathbf{D}\mathbf{s}_j\|^2$ with $\mathbf{s}_j = \mathbf{z}_j^{(t)} - \mathbf{z}_j^{(t-1)}$ for problem $j$ at iteration $t$.

Table 3: Study of stabilization techniques for learnt sparse coding. Denoising results in average PSNR with $\sigma = 25$ on BSD68.

| Method | Psnr (dB) | |
| --- | --- | --- |
| | D | C,D |
| BM3D [11] | 28.57 | |
| Sparse Coding (SC) | ✗ | ✗ |
| SC + Backtracking | 28.71 | 28.83 |
| SC + Spectral norm | 28.69 | 28.82 |
| SC + Barzilai-Borwein | 28.82 | **28.86** |

In our experiments, we observed that spectral normalization, backtracking, and Barzilai-Borwein step size were all effective to stabilize training. We have noticed that the spectral normalization impacts negatively the reconstruction accuracy, while the BB method tend to improve it by using larger stepsizes, at the expense of a larger computational cost. This is illustrated in Table 3 for a smoothed variant of sparse coding (we indicate with a crossmark when the algorithm diverges). In addition, we observe that the untied models brings a small boost in reconstruction accuracy.

## 4 Experiments

We consider three different tasks, illustrated with various combinations of regularization functions in order to demonstrate the wide applicability of our approach and its flexibility. A software package and additional details are provided in the supplementary material for reproducibility purposes.

**Image denoising.** For image denoising experiments, we use the standard setting of [63] with BSD400 [35] as a training set and on BSD68 as a test set. We optimize the parameters of our models using Adam [24] and also use the backtracking strategy described in Section 3.4 that automatically decreases the learning rate by a factor 0.5 when the training loss diverges. For the non-local models, we follow [26] and update the similarity matrices three times during the inference step. We use the parametrization with the $\mathbf{C}$ matrix for our patch-based experiments. We also combine our variance regularization with [26]. Additional training details and hyperparameters choices can be found in Appendix B. We report performance in terms of averaged PSNR in Table 4, and more detailed tables with additional results are available in Appendix C for pixel-level models, and for the patch-based models involving a dictionary $\mathbf{D}$. Our models based on non-local sparse approximations perform better than the competing deep learning models with the exception of [29] for $\sigma \geq 15$ with much fewer parameters. In addition, we also observed that our assymetric TV models are almost on par with BM3D while being significantly faster (see Appendix C for more details) with only a very small amount of parameters.

Table 4: **Grayscale denoising** on BSD68, training on BSD400 for all methods. Performance in terms of average PSNR. Tiny CNN is a CNN baseline with few parameters. See Appendix C for qualitative results.

| Method | Params | Noise Level ($\sigma$) | | | |
|---|---|---|---|---|---|
| | | 5 | 15 | 25 | 50 |
| Tiny CNN *(ours)* | 326 | 35.17 | 29.42 | 26.90 | 24.06 |
| Tiny CNN *(ours)* | 1200 | 36.47 | 30.36 | 27.70 | 24.60 |
| BM3D [11] | - | 37.57 | 31.07 | 28.57 | 25.62 |
| LSCC [34] | - | 37.70 | 31.28 | 28.71 | 25.72 |
| CSCnet [52] | 62k | 37.69 | 31.40 | 28.93 | 26.04 |
| GroupSC [26] | 68k | 37.95 | 31.71 | 29.20 | 26.17 |
| FFDNet [64] | 486k | N/A | 31.63 | 29.19 | 26.29 |
| DnCNN [63] | 556k | 37.68 | 31.73 | 29.22 | 26.23 |
| NLRN [29] | 330k | 37.92 | **31.88** | **29.41** | **26.47** |
| *Pixel-reconstruction* | | | | | |
| TV *symmetric* | 288 | 36.91 | 30.27 | 27.66 | 24.51 |
| TV *assymetric - extra-grad* | 480 | 37.30 | 30.76 | 28.24 | 25.32 |
| Laplacian *symmetric* | 288 | 35.17 | 28.42 | 26.14 | 23.70 |
| Laplacian *assymetric - extra-grad* | 480 | 35.20 | 28.46 | 26.39 | 23.77 |
| Bilateral - *extra-grad* | 146 | 36.75 | 29.89 | 27.20 | 23.72 |
| Bilateral TV - *extra-grad* | 146 | 36.94 | 30.46 | 27.78 | 24.52 |
| Non-local TV - *extra-grad* | 307 | 37.53 | 31.03 | 28.50 | 25.26 |
| Non-local Laplacian - *extra-grad* | 307 | 37.54 | 31.00 | 28.47 | 25.46 |
| *Patch-reconstruction* | | | | | |
| Sparse Coding (SC) | 68k | 37.84 | 31.46 | 28.90 | 25.84 |
| Sparse Coding + Variance | 68k | 37.83 | 31.49 | 29.00 | 26.08 |
| Sparse Coding + TV | 68k | 37.84 | 31.50 | 29.02 | 26.10 |
| Sparse Coding + TV + Variance | 68k | 37.84 | 31.51 | 29.03 | 26.09 |
| Non-local group | 68k | 37.95 | 31.69 | 29.19 | 26.19 |
| Non-local group + Variance | 68k | **37.96** | 31.70 | 29.22 | 26.28 |
| GroupSC + Variance | 68k | **37.96** | 31.75 | 29.24 | 26.34 |

**Compressed Sensing for fMRI.** Compressed Sensing for functional magnetic resonance imaging (fMRI) aims at reconstructing functional MR images from a small number of samples in the Fourier space. The corresponding inverse problem is

$$\min_{\mathbf{y} \in \mathbb{R}^n} \|\mathbf{A}\mathbf{y} - \mathbf{x}\|_2^2 + \lambda \Psi(\mathbf{y}), \tag{9}$$

where the degradation matrix is $\mathbf{A} = \mathbf{P}\mathcal{F}$, $\mathbf{P}$ is a diagonal binary sampling matrix for a given sub-sampling pattern, $\mathcal{F}$ is the discrete Fourier transform such that the observed corrupted signal $\mathbf{x}$ is in the Fourier domain, and $\Psi$ is a regularization function. This problem highlights the ability of our framework to handle both localized and non localized constraints. In our paper, we implemented two models revisiting some well studied priors for compressed sensing in an end-to-end fashion:

- *Pixel reconstruction* with total variation: we aim at solving the optimization for each node $\min_{\mathbf{y}_i \in \mathbb{R}} \|\mathbf{A}\mathbf{y} - \mathbf{x}\|_2^2 + \text{TV}_i(\mathbf{y})$. In the past, total variation has been widely used for MRI [30], often in combination with sparse regularization in the wavelet domain.
- *Patch encoding on a dictionary with sparse coding*: we solve a collection of optimization problems of the form $\min_{\mathbf{z}_i \in \mathbb{R}^n} \|\mathbf{A}\hat{\mathbf{y}}(\mathbf{z}) - \mathbf{x}\|_2^2 + \lambda\|\mathbf{z}_i\|_1$, with $\mathbf{y} = \frac{1}{n}\sum_j \mathbf{R}_j^\top \mathbf{D}\mathbf{z}_j$ the average of the overlapping patches. Some previous methods have explored dictionary-based reconstruction [47], but they were not investigated from a task-driven manner with end-to-end training.

In our experiments, we use the same setting as [53] for fair comparison: we train and test our models on the brain MRI dataset studied in that paper. Our models are trained separately for each sampling rate. We used the pseudo radial sampling for the matrix $\mathbf{P}$ similarly to the other methods. The reconstruction accuracy are reported in term of PSNR over the test set in Table 5. Our trainable model relying on a trainable TV prior performs surprisingly well given the conceptual simplicity of the prior. Also importantly, it runs significantly faster than all competing methods with a very small number of parameters. Furthermore, our trainable sparse coding method for fMRI gives strong performance and exceeds the state of the art for sampling rates larger than 30%. Note that architecture choices (patch and dictionary size) of our models are the same as for the denoising task, and we did not try to optimize them for the considered task, thus demonstrating the robustness of our approach.

Table 5: **Compressed sensing for fMRI** on the MR brain dataset using a pseudo radial sampling pattern. Performance comparisons in terms of PSNR (dB).

| Method | Params | 20 % | 30 % | 40% | 50% | Test time |
|---|---|---|---|---|---|---|
| TV [30] | - | 35.20 | 37.99 | 40.00 | 41.69 | 0.731s (cpu) |
| RecPF [59] | - | 35.32 | 38.06 | 40.03 | 41.71 | 0.315s (cpu) |
| SIDWT | - | 35.66 | 38.72 | 40.88 | 42.67 | 7.867s (cpu) |
| PANO [46] | - | 36.52 | 39.13 | 40.31 | 41.81 | 35.33s (cpu) |
| BM3D-MRI [13] | - | 37.98 | 40.33 | 41.99 | 43.47 | 40.91s (cpu) |
| ADMM-net [53] | - | 37.17 | 39.84 | 41.56 | 43.00 | 0.791s (cpu) |
| ISTA-net [62] | 337k | **38.73** | **40.89** | 42.52 | 44.09 | 0.143s (gpu) |
| CS-TV (ours) | 140 | 36.80 | 39.63 | 41.58 | 43.46 | 0.015s (gpu) |
| CS-Sparse coding (ours) | 68k | 37.80 | 40.50 | 42.46 | 44.16 | 0.213s (gpu) |
| CS-Sparse coding + Variance (ours) | 68k | 37.79 | 40.67 | **42.54** | **44.17** | 0.213s (gpu) |

Table 6: **Denoising with less data**. Results in terms of average PSNR(dB) on BSD68 with $\sigma = 15$. All the models are trained on a similar subset of BSD400 for fair comparaison.

| Method | Params | Training images | | | |
|---|---|---|---|---|---|
| | | 400 | 200 | 100 | 50 |
| DnCNN [63] | 556k | 31.73 | 31.65 | 31.47 | 31.23 |
| TV *extra-grad* | 480 | 30.75 | 30.72 | 30.67 | 30.66 |
| SC+Var | 68k | 31.49 | 31.49 | 31.47 | 31.40 |
| GroupSC+Var | 68k | **31.75** | **31.66** | **31.62** | **31.54** |

Table 7: **Dense stereo matching** fine-tuning on kitti2015 train set, performance reported on the kitti2015 validation set.

| Model | 3-px error (%) |
|---|---|
| PSMNet [8] | $2.14 \pm 0.04$ |
| PSMNet+TV 12 | $2.11 \pm 0.03$ |
| PSMNet+TV 24 | $2.11 \pm 0.04$ |
| PSMNet+TV *extra* | $\mathbf{2.10 \pm 0.03}$ |

**Dense Stereo Matching.** Our approach can be used to provide a generic regularization module that can easily be integrated into various neural architectures. We showcase its versatility by using it for deep stereo matching [51]. Given aligned image pairs, the goal is to compute disparity $\mathbf{d}$ for each pixel. Traditionally stereo matching is formulated as minimization of an energy function $E_{\text{data}}(\mathbf{d}) + \lambda E_{\text{smooth}}(\mathbf{d})$ where the data term, $E_{\text{data}}$ measures how well $\mathbf{d}$ agrees with the input image pairs, $E_{\text{smooth}}$ enforces consistency among neighboring pixels' disparities: TV is a commonly chosen regularizer. Recent deep learning methods tackle the problem as a supervised regression to estimate continuous disparity map given pairs of stereo views and ground truth disparity maps [8]. We propose to combine our smoothing TV block with a state-of-the-art deep learning model [8]. In practice, we combine our block with a pretrained model on the SceneFlow [36] dataset, and fine-tune the pretrained model on the kitti2015 [18] train set, following the training procedure described in [8]. We used the original implementation of [8] available online and did not change any hyperparameters. We report in Table 7 the performance on the validation set in term of 3 pixels error which counts predicted pixel as correct if the disparity deviates from the ground truth from 3 pixels or less. We ran the experiment 10 times for each model (with and without the TV regularization). We observed that our TV block introduces very few additional parameters and consistently boosts performances.

**Training with few examples.** We conducted denoising experiments with less training data and report corresponding results in Table 6. We use the code released by the authors for training DnCNN with less data. Very interestingly the gap between our best model and CNN-based models increases when decreasing the size of the training set. We believe that this is an appealing feature, particularly relevant for applications in medical imaging or microscopy where the amount of training data can be very limited.

## 5   Discussion

We have presented a general framework based on non-cooperative games to train end-to-end imaging priors. Our experiments demonstrate the flexibility and the effectiveness of our approach on diverse tasks ranging from image denoising to fMRI reconstruction and dense stereo matching. Beyond image processing, we believe that the issue of interpretability is important. We consider models with a clear mathematical description of the decision function they produce. As a by-product, our models are also more parameter efficient than classical deep learning models. We believe that these are important steps to build systems that should not be seen as black boxes anymore, that produce explanable decisions, and that do not require training a system for days on a huge corpus of annotated data. These are important questions, which we are planning to address explicitly in the future.

## Acknowledgments and Disclosure of Funding

This work was funded in part by the French government under management of Agence Nationale de la Recherche as part of the "Investissements d'avenir" program, reference ANR-19-P3IA-0001 (PRAIRIE 3IA Institute). JM and BL were supported by the ERC grant number 714381 (SOLARIS project) and by ANR 3IA MIAI@Grenoble Alpes (ANR-19-P3IA-0003). JP was supported in part by the Louis Vuitton/ENS chair in artificial intelligence and the Inria/NYU collaboration. Finally, this work was granted access to the HPC resources of IDRIS under the allocation 2020-AD011011252 made by GENCI.

## Broader Impact

Our main field of application is in image processing, with a focus on image restoration and reconstruction, whose benefits for society are clear and well established, even though a misuse of such a technology is of course possible; for instance, the same total variation penalty may be used in medical imaging, for personal photography, in astronomical imaging, or for restoring images produced by military devices. More specifically, our paper is addressing the issue of interpretability of neural networks, by considering models admitting a functional (mathematical) description of their decision functions, and with less parameters than classical deep learning models. As we mentioned in the discussion section, we believe that these are first steps to build systems producing explanable decisions, and that are more data and energy efficient. These are important issues going beyond image processing, which we would like to address in future work.

## Footnotes

*Inria, École normale supérieure, CNRS, PSL Research University, 75005 Paris, France

†Inria, Univ. Grenoble Alpes, CNRS, Grenoble INP, LJK, 38000 Grenoble, France

[3]We employ a debiasing dictionary $\mathbf{W} \neq \mathbf{D}$ to improve the quality of the reconstructions. Debiaising is commonly used when dealing with $\ell_1$ penalty which is known to shrink the coefficients $\mathbf{Z}$ too much.

[4]`torch.nn.functionnal.conv2D_transpose` on PyTorch [44].

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
