[Supplementary Material]

# Appendix

This supplementary material is organized as follows: In Section A, we discuss additional priors that were not presented in the main paper, but which are in principle compatible with our framework, and we provide more details about potential games. In Section B, we provide implementation details that are useful to reproduce the results of our paper (note that the code is also provided). In Section C, we present additional quantitative results and additional results regarding inference speed of our models that were not included in the main paper for space limitation reasons. Finally, in Section D, we present additional qualitative results (which require zooming on a computer screen).

## A   Discussion on Models and Priors

### A.1   Additional Priors

Our framework makes it possible to handle models of the form:

$$h_j(\mathbf{Z}) = h_{\boldsymbol{\theta}}(\mathbf{P}_j\mathbf{x}_j, \mathbf{z}_j) + \lambda \sum_{k=1}^{r} \phi_k(L_{k,j}(\mathbf{Z})), \tag{10}$$

where $\phi_k$ is a simple convex function that admits a proximal operator in closed form, and $L_{k,j}$ is a linear operator. In the main paper, several regularization functions have been considered, including the total variation, variance reduction, or non-local group regularization penalties. Here, we would like to mention a few additional ones, which are in principle compatible with our framework, but which we did not investigate experimentally. In particular, two of them may be of particular interest, and may be the topic of future work:

- the regularization $\lambda\|\mathbf{H}^\top\mathbf{z}_j\|_1$, where $\mathbf{H}$ is a matrix, may correspond to several settings. The matrix $\mathbf{H}$ may be for instance a wavelet basis, or may by learned, corresponding then to the penalty used in the analysis dictionary learning model from the paper "The cosparse analysis model and algorithms" of Nam et al., 2013.
- the regularization $\lambda\phi(\mathbf{H}^\top\mathbf{z}_j)$ where $\phi$ is a smooth function is closely related to the model introduced in [27], and to the Field of experts model of Roth and Black from the 2005 paper "Fields of Experts: A Framework for Learning Image Priors", even though the functions used in these other works are not convex.

### A.2   Potential Games

A potential game is a non-cooperative convex game whose Nash equilibria correspond to the solutions of a convex optimization problem. We will now consider problems of the form (10), and show that all penalties that admit some symmetry are in fact potential games. Assuming the functions $\phi_k$ to be smooth for simplicity, optimality conditions for the convex problems (10) are, for all $j = 1, \ldots, m$:

$$\nabla_{\mathbf{z}_j} h_{\boldsymbol{\theta}}(\mathbf{P}_j\mathbf{x}_j, \mathbf{z}_j) + \lambda \sum_{k=1}^{r} \nabla_{\mathbf{z}_j}\tilde{\phi}_{k,j}(\mathbf{Z}) = 0, \quad \text{with} \quad \tilde{\phi}_{k,j}(\mathbf{Z}) = \phi_k(L_{k,j}(\mathbf{Z})). \tag{11}$$

Let us now assume the following symmetry condition such that if problem $l$ involves a variable $\mathbf{z}_j$ through a function $\tilde{\phi}_{k,l}(\mathbf{Z})$, then problem $j$ also involves the same term. Based on this assumption, we may define the potential function

$$V(\mathbf{Z}) := \sum_{j=1}^{m} \left( h_{\boldsymbol{\theta}}(\mathbf{P}_j\mathbf{x}_j, \mathbf{z}_j) + \frac{\lambda}{2} \sum_{k=1}^{r} \tilde{\phi}_{k,j}(\mathbf{Z}) \right).$$

The partial derivative of this potential function with respect to $\mathbf{z}_j$ is then

$$\nabla_{\mathbf{z}_j} h_{\boldsymbol{\theta}}(\mathbf{P}_j\mathbf{x}_j, \mathbf{z}_j) + \frac{\lambda}{2} \sum_{l=1}^{m} \sum_{k=1}^{r} \nabla_{\mathbf{z}_j}\tilde{\phi}_{k,l}(\mathbf{Z}) = \nabla_{\mathbf{z}_j} h_{\boldsymbol{\theta}}(\mathbf{P}_j\mathbf{x}_j, \mathbf{z}_j) + \frac{\lambda}{2} \sum_{l=1}^{m} \sum_{k \in \mathcal{N}_{j,l}} \nabla_{\mathbf{z}_j}\tilde{\phi}_{k,l}(\mathbf{Z}),$$

where $\mathcal{N}_{j,l}$ is the set of functions $\tilde{\phi}_{k,l}$ involving variable $\mathbf{z}_j$. The previous gradient can then be simplified into

$$\nabla_{\mathbf{z}_j} h_{\boldsymbol{\theta}}(\mathbf{P}_j\mathbf{x}_j, \mathbf{z}_j) + \frac{\lambda}{2} \sum_{j=1}^{r} \nabla_{\mathbf{z}_j}\tilde{\phi}_{k,l}(\mathbf{Z}) + \frac{\lambda}{2} \sum_{l \neq j} \sum_{k \in \mathcal{N}_{j,l}} \nabla_{\mathbf{z}_j}\tilde{\phi}_{k,l}(\mathbf{Z}).$$

Since the symmetry condition can be expressed as $\sum_{j=1}^{r} \tilde{\phi}_{k,l}(\mathbf{Z}) = \sum_{l \neq j} \sum_{k \in \mathcal{N}_{j,l}} \tilde{\phi}_{k,l}(\mathbf{Z})$, the condition $\nabla V(\mathbf{Z}) = 0$ is then equivalent to (11). Note that we have assumed the functions $\phi_k$ to be smooth for simplicity, but a similar reasoning can be conducted for non-smooth functions, by using the concept of subgradients.

**Examples of potential games.**
- the $\ell_1$-norm: with $r = 1$ and $\tilde{\phi}_{1,j} = \|\mathbf{z}_j\|_1$, since problem $j$ does not involve any variable $\mathbf{z}_l$ for $l \neq j$;
- Symmetric TV / Laplacian: problem $j$ may involve a variable $\mathbf{z}_l$ through a term $a_{j,l}\|\mathbf{z}_j - \mathbf{z}_l\|_1$. Then, problem $l$ involves the same term $a_{l,j}\|\mathbf{z}_j - \mathbf{z}_l\|_1$ under the condition $a_{j,l} = a_{l,j}$.
- Symmetric non local group with $r = p$ and $\tilde{\phi}_{k,j} = \lambda_k \|[\sqrt{a_{j,1}}\mathbf{z}_1(k), \ldots, \sqrt{a_{j,m}}\mathbf{z}_m(k)]^\top\|_2$. Under the condition of symmetric weights $a_{j,l} = a_{l,j}$, we obtain again a potential game.

Potential games are appealing as they provide guarantees about the existence of Nash equilibria without requiring optimizing over a compact set. Yet, we have found that allowing non-symmetric weights often performs better. This is illustrated in Table A1 for a simple denoising experiment.

Table A1: **Symmetric vs assymmetric** grayscale denoising on BSD68, training on BSD400 for all methods. Performance is measured in terms of average PSNR.

| Method | Params | Noise Level ($\sigma$) | | | |
| --- | --- | --- | --- | --- | --- |
| | | 5 | 15 | 25 | 50 |
| TV *symmetric* | 72 | 36.08 | 30.21 | 27.58 | 24.74 |
| TV *assymetric - extra-grad* | 480 | 37.30 | 30.76 | 28.24 | 25.32 |
| Laplacian *symmetric* | 72 | 34.88 | 28.14 | 25.90 | 23.45 |
| Laplacian *assymetric - extra-grad* | 480 | 35.20 | 28.46 | 26.39 | 23.77 |
| Non-local group - *symmetric* | 68k | 37.94 | 31.67 | 29.17 | 26.16 |
| Non-local group - *assymetric* | 68k | 37.95 | 31.69 | 29.20 | 26.19 |

# B  Implementation Details and Reproducibility

## B.1  Training Details

For the training of patch-based models for denoising, we randomly extract patches of size $56 \times 56$ whose size equals the window size used for computing non-local self-similarities; whereas we train pixel level models on the full size images. For fMRI experiments we also trained the models on the full sized images. We apply a mild data augmentation (random rotation by $90°$ and horizontal flips). We optimize the parameters of our models using ADAM [24].

The learning rate is set to $6 \times 10^{-4}$ at initialization and is sequentially lowered during training by a factor of $0.35$ every 80 training steps, in the same way for all experiments. Similar to [52], we normalize the initial dictionary $\mathbf{D}_0$ by its largest singular value as explained in the main paper in Section 3.4. We initialize the dictionary $\mathbf{C}, \mathbf{D}$ and $\mathbf{W}$ with the same dictionary obtained with an unsupervised dictionary learning algorithm (using SPAMS library).

We have implemented the backtracking strategy described in Section 3.4 of the main paper for all our algorithms, which automatically decreases the learning rate by a factor $0.8$ when the loss function increases too much on the training set, and restore a previous snapshot of the model. Divergence is monitored by computing the loss on the training set every 10 epochs. Training the non-local models for denoising are the longer models to train and takes about 2 days on a Titan RTX GPU. We summarize the chosen hyperparameters for the experiments in Table A2.

# C  Additional Quantitative Results

## C.1  Inference speed

In Table A3 we provide a comparison of our TV models in terms of speed with BM3D for grayscale denoising on the BSD68 dataset. For fair comparison, we reported computation time both on gpu and cpu.

## C.2  Image denoising

We provide additional results for grayscale denoising with different variations of the prior introduced in the main paper, as well as combination of different priors. We reported performances for gray

Table A2: Hyper-parameters chosen for every task.

| Experiment | Gray denoising (patch) | Gray denoising (pixel) | fMRI |
|---|---|---|---|
| Patch size | 9 | - | 9 |
| Dictionary size | 256 | - | 256 |
| Nr epochs | 300 | 300 | 150 |
| Batch size | 32 | 32 | 1 |
| $K$ iterations | 24 | 24 | 24 |
| Middle averaging | ✓ | ✓ | - |
| Correlation update frequency $f$ | 1/6 | 1/12 | - |

Table A3: Inference speed for image denoising.

| | Params | Psnr | Speed |
|---|---|---|---|
| BM3D [11] | - | 25.62 | 7.28s (cpu) |
| TV assymetric | 240 | 24.93 | 0.014s (gpu) / 0.18s (cpu) |
| TV assymetric (extra) | 480 | 25.32 | 0.021s (gpu) / 0.28s (cpu) |

denoising in Table A4 for the pixel based models, and in Table A5 for the patch based models. In Table A4 *untied* $\kappa$ denotes when we used a different set of learned parameters $\kappa$ at each stage of the refinement step of the similarity matrix for the non-local models.

Table A4: **Pixel level** grayscale denoising on BSD68, training on BSD400 for all models. Performance is measured in terms of average PSNR.

| Method | Params | Noise Level ($\sigma$) | | | |
|---|---|---|---|---|---|
| | | 5 | 15 | 25 | 50 |
| BM3D [11] | - | 37.57 | **31.07** | **28.57** | **25.62** |
| Tiny CNN | 326 | 35.17 | 29.42 | 26.90 | 24.06 |
| Tiny CNN | 1200 | 36.47 | 30.36 | 27.70 | 24.60 |
| TV *symmetric* | 288 | 36.08 | 30.21 | 27.58 | 24.74 |
| TV *symmetric - extra-grad* | 144 | 37.02 | 30.33 | 27.82 | 24.81 |
| TV *assymetric-* | 240 | 36.83 | 30.49 | 27.99 | 24.93 |
| TV *assymetric - extra-grad* | 480 | 37.30 | 30.76 | 28.24 | 25.32 |
| Laplacian *symmetric* | 288 | 34.88 | 28.14 | 25.90 | 23.45 |
| Laplacian *symmetric - extra-grad* | 144 | 33.87 | 28.14 | 25.91 | 23.45 |
| Laplacian *assymetric* | 240 | 35.20 | 28.48 | 26.17 | 23.78 |
| Laplacian *assymetric - extra-grad* | 480 | 35.20 | 28.46 | 26.39 | 23.77 |
| Non-local TV *assymmetric* | 154 | 37.25 | 30.86 | 28.28 | 25.42 |
| Non-local TV *assymmetric* (untied $\kappa$) | 235 | 37.12 | 31.01 | 28.37 | 25.24 |
| Non-local TV *assymmetric - extra-grad* | 226 | **37.83** | 30.98 | 28.34 | 25.31 |
| Non-local TV *assymmetric - extra-grad* (untied $\kappa$) | 307 | 37.53 | 31.03 | 28.50 | 25.26 |
| Non-local Laplacian *assymmetric* | 154 | 37.31 | 30.75 | 28.33 | 25.15 |
| Non-local Laplacian *assymmetric* (untied $\kappa$) | 235 | 37.53 | 31.01 | 28.37 | 25.47 |
| Non-local Laplacian *assymmetric - extra-grad* | 226 | 37.51 | 30.99 | 28.34 | 25.13 |
| Non-local Laplacian *assymmetric - extra-grad* (untied $\kappa$) | 307 | 37.54 | 31.00 | 28.47 | 25.46 |
| Bilateral | 74 | 36.76 | 29.89 | 27.16 | 23.97 |
| Bilateral TV | 74 | 36.60 | 29.82 | 27.23 | 24.00 |
| Bilateral - *extra-grad* | 146 | 36.75 | 29.89 | 27.20 | 23.72 |
| Bilateral TV - *extra-grad* | 146 | 36.94 | 30.46 | 27.78 | 24.52 |

# D   Additional Qualitative Results

Finaly, we show qualitative results for grayscale denoising in Figures A3, A4.

Table A5: **Patch level** grayscale denoising on BSD68, training on BSD400 for all methods. Performance is measured in terms of average PSNR.

| Method | Params | Noise Level ($\sigma$) | | | |
|---|---|---|---|---|---|
| | | 5 | 15 | 25 | 50 |
| BM3D [11] | - | 37.57 | 31.07 | 28.57 | 25.62 |
| LSCC [34] | - | 37.70 | 31.28 | 28.71 | 25.72 |
| CSCnet [52] | 62k | 37.69 | 31.40 | 28.93 | 26.04 |
| FFDNet [64] | 486k | N/A | 31.63 | 29.19 | 26.29 |
| DnCNN [63] | 556k | 37.68 | 31.73 | 29.22 | 26.23 |
| NLRN [29] | 330k | 37.92 | **31.88** | **29.41** | **26.47** |
| GroupSC [26] | 68k | 37.95 | 31.71 | 29.20 | 26.17 |
| Sparse Coding + Barzilai-Borwein | 68k | 37.85 | 31.46 | 28.91 | 25.84 |
| Sparse Coding + Variance | 68k | 37.83 | 31.49 | 29.00 | 26.08 |
| Sparse Coding + TV | 68k | 37.84 | 31.50 | 29.02 | 26.10 |
| Sparse Coding + TV + Variance | 68k | 37.84 | 31.51 | 29.03 | 26.09 |
| Sparse Coding + TV + Variance + Barzilai-Borwein | 68k | 37.86 | 31.52 | 29.04 | 26.04 |
| Non-local group - *symmetric* | 68k | 37.94 | 31.67 | 29.17 | 26.16 |
| Non-local group - *assymetric* | 68k | 37.95 | 31.69 | 29.20 | 26.19 |
| Non-local group - *assymetric* + TV | 68k | 37.96 | 31.71 | 29.22 | 26.26 |
| Non-local group - *assymetric* + Variance | 68k | 37.96 | 31.70 | 29.23 | 26.28 |
| Non-local group - *assymetric* + Variance + TV | 68k | 37.95 | 31.71 | 29.24 | 26.30 |
| GroupSC + Variance | 68k | **37.96** | 31.75 | 29.24 | 26.34 |

Figure A3: Grayscale denoising for 4 images from the BSD68 dataset. Best seen by zooming on a computer screen.

| | | |
|---|---|---|
| ground truth | noisy | BM3D |
| DnCNN | SC (ours) | Non local (ours) |

| | | |
|---|---|---|
| ground truth | noisy | BM3D |
| DnCNN | SC (ours) | Non local (ours) |

| | | |
|---|---|---|
| ground truth | noisy | BM3D |
| DnCNN | SC (ours) | Non local (ours) |

| | | |
|---|---|---|
| ground truth | noisy | BM3D |
| DnCNN | SC (ours) | Non local (ours) |

Figure A4: Results of our patch level models for grayscale denoising for 4 images from the BSD68 dataset. Best seen by zooming on a computer screen.