[Reviews · NeurIPS 2020]

Review 1

Summary and Contributions: This paper aims to develop a non-cooperative game setting whose aim is to simultaneously design signal priors and corresponding optimization methods, all within the context of a neural type architecture trainable by automatic differentiation. The authors consider both gradient l1 (TV) an l2 (Laplacian) type regularization methods, as well as "standard" l1 and l2 methods, for image restoration tasks, and demonstrate the efficacy of their approaches via PSNR values; actual images are shown in the supplement. UPDATE: The authors commented their framework is actually a bit broader.

Strengths: The framework proposed is fairly flexible, and the results arising from it here are compelling. The overall learning formulation being posed as a cooperative game seems to be novel.

Weaknesses: The exposition was, in my opinion, a bit too "high level" throughout. For example, despite this being fundamentally an algorithmic paper that uses a standard prototype algorithm, it was correspondingly difficult to parse specifically what that algorithm was. While I appreciate the authors including source code, I personally would prefer that pseudocode for the algorithm be clearly spelled out in the paper too. Along related lines, the exposition went back and forth between a more general, high level, abstract formulation of the overall problem, and a more specific image patch-based example. Maybe the "Tricks of the Trade" could be moved somewhere closer to the beginning of the paper where the more general framework was discussed. UPDATE: The former comments weren't addressed by the authors. In my opinion, PSNR values by themselves don't really provide an overly compelling justification for image-based methods. It would be much better to actually see the differences between the competing methods visually, in the main body of the paper. UPDATE: The authors pledged to include some additional justification in a revision. UPDATE: I still hold that this is a unique perspective worth exploring, but could/should be more clearly exposed.

Correctness: There are no significant analytical claims here. The methodology seems correct at a high level, but more specific details are missing outside of the included simulation code.

Clarity: Some restructuring, along the lines of the comments above, would improve the readability in my opinion.

Relation to Prior Work: The paper was well positioned with respect to existing prior works.

Reproducibility: Yes

Additional Feedback: N/A


Review 2

Summary and Contributions: This contribution proposes a framework to learn specific priors for convex optimization problems. This framework is a generalization of the bi-level formulation for the unrolled optimization of [36]. Instead of averaging the effect of the regularization over a sample, a non cooperative convex game is used. The authors demonstrate their framework on several classic priors: TV, l1, non-local TV,... The paper also provides several tricks to train these special networks. The advantages of this technique are twofolds. First the obtained models are more interpretable as the forward pass is equivalent to solve well-known convex optimization problem. Second, the amount of computation and parameters is much smaller than classical neural net. However, they compromise a bit in term of performance. The experiment section is convincing.

Strengths: * Relatively novel * Technically accurate * Meaningful contribution for Green AI and interpretability

Weaknesses: * In several practical cases, the non-cooperative game can be solved by minimizing the sum of all the convex terms, i.e. the setting called a potential game. In this case we recover a classical convex optimization problem. If I understand correctly, we then recover the classical setting or eq.1? It also seems to me that all priors and problems within the context of this paper are convex. (I am not sure about the dictionary learning case for this particular framework.) If this is the case, the generalization to non-cooperative game seems a bit artificial to me. I might also be wrong. Could you please elaborate? # Post review comment: I am still not convinced by the author response and I am not sure that I fully understand this central point in the paper. Hence my score as well as my confidence score.

Correctness: I believe it is. However, I have not checked the Appendix or the code.

Clarity: I definitely feel the Neurips page limit. Overall, I believe the authors did a relatively good job. I would emphasize a bit more the difference between the non-cooperative game setup and eq. 1 as this part was not very clear to me.

Relation to Prior Work: Overall yes. I found a bit confusing that sometimes the original (convex optimization) paper is cited although I would expect the unrolled neural network version of it. For example after eq 1.

Reproducibility: Yes

Additional Feedback: Do you need fewer samples? Maybe you should mention that this work goes in the direction of green AI? # Post review edit: Thank you for the additional experiment. I believe it is very insideful... Line 69: Eq 1. you use the symbol \in and not =, is p>m?, is there multiple solutions? Line 75: add a reference? Line 76-79: this is a long sentence... Line 111: at some point the patches overlap. Maybe that should be mentioned before. Line 113: I think P_j should *not* be transposed here Line 119: I think it should not be x_j but x in the norm. Line 132-133 Total variation: this is fairly different from the TV from Chambolle. Maybe mention that the gradient is done by the difference of the z. What is \mathcal{N}? Is it the same as \mathcal{N}_j, you probably share the weights, maybe mention it. Line 132: No caption in the table? Line 144: convolutional sparse coding-> Convolutional Sparse Coding Line 163: I think you should specify that h is convex for z given the other parameters fixed. Line 177: and then to use? In Table 1: GD is not defined, I guess you mean Gradient


Review 3

Summary and Contributions: This paper considers learning for a representation of the image signal with domain-specific priors (through regularizers, either smooth or non-smooth) as part of the end-to-end training pipeline for final prediction tasks. By incorporating informative priors, extensive empirical results demonstrate its effectiveness compared to existing methods with much larger number of parameters. The optimization for the encoding step is modeled as a non-cooperative convex game and first-order algorithm for solving it is unrolled for computing updates to the parameter using auto-differentiation. Heuristic tricks are also discussed for improving practical performance.

Strengths: Empirical evaluation of the paper is quite thorough and strong. The setup under study could see a lot of applications in a diverse range of image processing applications.

Weaknesses: The proposed framework draws upon existing literature on unrolling optimization for end-to-end training; some of the extensions/generalizations seem relatively straightforward. [Discussion phase addition: I would like to thank the authors for their response and the clarification. Overall I think the presentation of the paper could be improved but otherwise think it's a good submission and would like to leave my view unchanged. ]

Correctness: The method and claims look correct.

Clarity: Paper is mostly clearly written, well-motivated and easy to follow.

Relation to Prior Work: The related work section adequately surveys the prior works.

Reproducibility: Yes

Additional Feedback: \item Notation: In equation (4), or tracing back to equation (3), no $\theta$ dependence is explicitly written, which makes the decision variable $\theta$ in the optimization problem a bit implicit. \item From the discussion in Section 3.2, it's elaborated how one would learn $z_\theta^*$ so the optimization w.r.t $\theta$ part is clear, but the update rule for $W$ is never mentioned - is it updated through backpropagation along with $\theta$? I ask because it looks like $a_{j,k}$ depends on $W$ through $\hat{y}$, which makes $z_j^*(x)$ a function of both $\kappa$ and $W$. But looking at the last column of the table on page 4, for the Non-local total variation for example, only $\kappa$ is considered as the model parameter $\theta$ in the (convex?) regularization term $\psi_\theta$, which is slightly confusing.


Review 4

Summary and Contributions: In this paper the authors present a framework for image denoising, basically by designing a neural network based on optimization algorithms, but with more general prior functions than the predecessors, and including in the cost function a more general approach. The paper is well written in general, it includes examples of each type of functions like models, loss functions, regularization terms, etc. I'm quite far from being an expert in this filed, so it took me a while to understand the big picture, what the whole method was doing in the end. The introduction is correct, and so are the next sections, and I got the feeling that I was following step by step the procedure, but yet in the end it was hard for me to understand what the method was doing. The main contribution is the the general framework, with a solid ground on models/optimization techniques, that allows to train an interpretable model with few parameters, obtaining very good results on image processing tasks.

Strengths: Although there are no theoretical results in the paper, the theoretical ground is a strength, in the sense of the deductions, models, and methods used to get to the proposed framework. This is not a paper trying to explain theoretically why neural networks work, but it is neither just a model that works, without proper justification. The experimental results are promising as well, with fewer parameters than other methods, and the interpretability that comes from the optimization algorithms used as inspiration. To the best of my knowledge, the generality of the model within this framework is new, and it is definitively relevant to the NeurIPS community.

Weaknesses: The weakness is the lack of theoretical guarantees, although its not a weakness of the paper itself, but of the family of methods in general. This being said, the general theoretical grounding, in terms of optimization for instance, is sound.

Correctness: The deductions made through the paper are standard, and they're correct. The empirical methodology is correct as well. The comparisons seem to be fair.

Clarity: The paper is well written in general, with the observations made in point 1.

Relation to Prior Work: The bibliography in general is abundant. The pior work is clearly presented, and the puntual differences are stated. I would use the opportunity in that part to explain the differences more globally.

Reproducibility: Yes

Additional Feedback: Minor comments: - Line 89. It's weird that just after saying "a more general point of view", it assumes a patch structure. I know what you meant, it's just that the placing is weird. - Line 177 "and then us to use auto" - Line 197 "typically be adapted handle" -> "typically be adapted to handle" ------------------------- I I've read the authors' response, but my review doesn't change.

[Author Response · NeurIPS 2020]

We thank the reviewers for their constructive comments. We provide our answers below.

**Presentation**: if the paper is accepted, we will improve our presentation, notably by better explaining "the big picture"
and what our method does (R4); including detailed pseudo code for our algorithm (R1); emphasizing the difference
between the introduced game setup and the standard bi-level formulation (R2); discussing data modalities without patch
structure to which our approach is applicable, and distinguishing better our general methodology and its aspects specific
to images (R1).

**R1:** "The authors consider both gradient l1 an l2 type regularization, as well as "standard" l1 and l2 methods [..]".
In fact, our framework also handles a broader range of regularizers including non-local variants, sparse coding, and
non-local group sparsity. "PSNR values by themselves don't really provide an overly compelling justification". We
agree with the reviewer. If the paper is accepted we will include at least some qualitative examples in the main body of
the paper and additional ones in the supplemental material. Finally, we also plan to include SSIM scores. See Table 1
below for an overview.

**R2:** " [..] all priors and problems within the context of this paper are convex". This is correct : we focus on convex
games, i.e., each objective function $h_j$ is convex wrt. $\mathbf{Z}$. "[..] the generalization to non-cooperative game seems a
bit artificial to me". We can recover a classical optimization problem by summing convex regularizers. However an
interesting insight from our experiments is that it is often beneficial to consider non-potential games instead. Please
refer to Table 1 in our paper and Table A2 in the appendix for more details regarding this point. "Do you need
fewer samples?". To answer this we have conducted additional experiments and report the results in Table 2 below.
Interestingly, our method outperforms DnCNN for all numbers of training images used in the experiments, the gap
in performance increasing significantly as the number of images decreases. This may be particularly interesting in
applications where the amount of training data is very limited (for example in medical imaging). A more detailed
study will be included if the paper is accepted. *Additional feedback*: we will correct typos and answer the rest of the
questions below. Line 69: we do not have necessarily $p > m$ even though it is generally the case for sparse coding.
"Is there multiple solutions". This could be the case (if $h$ is not strongly convex). Line 113: $\mathbf{P}^\top$ denotes the operator
which places patches on the reconstructed image. Line 119: Thank you, this will be corrected in the final version. Line
132-133:" this is fairly different from the TV from Chambolle". In the case of TV, we are in the pixel-level setting,
hence $\mathbf{z}_j$ directly models the underlying value of the observed pixel $\mathbf{x}_j$. So our TVs are in fact similar (we consider
here the anisotropic version). "What is $\mathcal{N}$? Is it the same as $\mathcal{N}_j$". Yes, the weights are shared. "I guess you mean
Gradient". Yes we do.

**R3:** "[..] some of the extensions/generalizations seem relatively straightforward". We agree that our method makes
the training of the studied priors relatively easy thanks to smoothing techniques and game encoding, and actually we
believe that this is one of the strengths. To the best of our knowledge, no prior works proposed unrolled TV models
and only one heuristic approach was proposed for group sparsity [28]. *Additional feedback*: "No $\theta$ dependence is
explicitly written [..]". Thank you, we will clarify the notation. "[..] the update rule for $\mathbf{W}$ is never mentioned [..]". For
patch-based models we employ a debiasing dictionary $\mathbf{W}$ to improve the quality of the reconstructions. Debiasing is
commonly used when dealing with $\ell_1$ penalty which is known to shrink the coefficients $\mathbf{Z}$ too much. In our method,
$\mathbf{W}$ is learned through backpropagation. However, when dealing with pixel-based regularizers (including TV), we are
in the setup where $q = 1 \times 1$ and $p = 1$ and the $\mathbf{W}$ matrix boils down to a single scalar coefficient. Empirically, this
coefficient does not impact performance significantly so it was neglected in our implementation. We will clarify this
point in the final version. "[..] it looks like $a_{j,k}$ depends on $\mathbf{W}$ through $\hat{\mathbf{y}}$". This is the case when the graph is non-local.
We admit that the table can be confusing so it will be updated in the final version.

**R4:** "The weakness is the lack of theoretical guarantees, although its not a weakness of the paper itself, but of the family
of methods in general". Our work does have theoretical in the case of a potential games. However, we did not manage
to obtain convergence guarantees when considering a general convex game. Proving the convergence of the forward
inference algorithm amounts to showing the monotonicity of the $H$ operator, which turns out to be very challenging
(very limited results on this type of problem are available in the literature). This is an interesting direction for future
research. Thank you for reporting typos which will be corrected in the final version.

Table 1: Denoising results in terms of average PSNR(dB)/SSIM on BSD68.

| Method | Noise level $\sigma$ | | |
| | 15 | 25 | 50 |
| --- | --- | --- | --- |
| DnCNN[70] | 31.73/0.8907 | 29.23/0.8278 | 26.23/0.7189 |
| TV | 30.75/0.8614 | 28.24/0.7903 | 25.32/0.6619 |
| SC+Var | 31.49/0.8885 | 29.00/0.8234 | 26.08/0.7088 |
| Group+Var | **31.75/0.8970** | **29.24/0.8341** | **26.34/0.7310** |

Table 2: Denoising ($\sigma = 15$) with smaller training sets. Results in terms of average PSNR(dB) on BSD68.

| Method | Params | Training images (BSD400) | | | |
| | | 400 | 200 | 100 | 50 |
| --- | --- | --- | --- | --- | --- |
| DnCNN [70] | 556k | 31.73 | 31.65 | 31.47 | 31.23 |
| TV | 480 | 30.75 | 30.72 | 30.67 | 30.66 |
| SC+Var | 68k | 31.49 | 31.49 | 31.46 | 31.40 |
| Group+Var | 68k | **31.75** | **31.66** | **31.62** | **31.54** |

[Meta-Review · NeurIPS 2020]

This paper proposes a framework for building trainable variants of image priors and recipes to facilitate their training. The reviewers appreciate the flexibility of the framework, the viewpoint of noncooperative game to formulate the problem, and promising empirical results. Overall, reviewers are positive about this work, and based on their given score, I recommend accept. However, please note that all reviewers believed the presentation can improve. Please apply reviewers' feedback to the final draft. In particular, R1 has concrete suggestions that are easy to incorporate.